# The combination of asymmetric hydrogenation of olefins and direct reductive amination

Shuai Yuan[1], Guorui Gao[2], Lili Wang[1], Cungang Liu[1], Lei Wan[1], Haizhou Huang[1]*, Huiling Geng[1]* & Mingxin Chang ⓘ [1]*

Asymmetric hydrogenation (AH) and direct reductive amination (DRA) are both efficient transformations frequently utilized in industry. Here we combine the asymmetric hydrogenation of prochiral olefins and direct reductive amination of aldehydes in one step using hydrogen gas as the common reductant and a rhodium-Segphos complex as the catalyst. With this strategy, the efficiency for the synthesis of the corresponding chiral amino compounds is significantly improved. The practical application of this synthetic approach is demonstrated by the facile synthesis of chiral 3-phenyltetrahydroquinoline and 3-benzylindoline compounds.

[1] Shaanxi Key Laboratory of Natural Products & Chemical Biology, College of Chemistry & Pharmacy, Northwest A&F University, 22 Xinong Road, Yangling 712100 Shaanxi, China. [2] College of Chemistry, Chemical Engineering and Materials Science, Collaborative Innovation Center of Functionalized Probes for Chemical Imaging in Universities of Shandong, Shandong Normal University, 88 Wenhuadong Road, 250014 Jinan, China. *email: huanghai30@163.com; genghuiling@nwsuaf.edu.cn; mxchang@nwsuaf.edu.cn

Since the manufacture of L-dopa was realized as the first successful industrial-scale asymmetric catalytic process[1], asymmetric hydrogenation (AH) has become the main driving force for asymmetric catalysis[2–7], and the most frequently utilized homogeneous enantioselective catalytic tranformation in large scale[8–17]. The striding progress in AH research was evinced by Knowles[18] and Noyori[19] winning the Nobel prize. AH bears a few outstanding merits: first of all hydrogen gas as the reductant offers 100% atom economy; at the same time, there is a vast library of readily available chiral ligands; more importantly, the catalytic activity is prominent. The catalyst loading can be as low as 0.00002 mol%[20]. As a result, many important fine chemicals and active pharmaceutical ingredients (APIs) are manufactured via AH of C=C, C=O and C=N bonds, including L-menthol (the most manufactured chiral compound), metolachlor (the best-selling chiral herbicide), dextromethorphan, fluoxetine and sitagliptin[2–17].

At the other hand, direct reductive amination (DRA), a one-pot procedure for the construction of C–N bond in which the mixture of carbonyl compound and nitrogen-containing compound is subjected to a reducing agent with $H_2O$ as the sole by-product, is a key transformation in organic chemistry[21–41]. It has been developed into one of the most practical methods for the synthesis of amino pharmaceutical compounds. Common reducing agents include $H_2$, borohydride, formate, silane, isopropyl alcohol and Hantzsch esters. Transition metal-catalyzed DRA using molecular hydrogen as the reducing reagent is highly sustainable in terms of reaction waste control and atom-economy. As both AH and DRA share the same reductant, $H_2$, we propose to combine these two reactions to make the procedure for the synthesis of chiral amino compounds more concise and proficient (Fig. 1). To achieve this purpose, two major challenges have to be addressed: the discovery of a catalytic system which can perform reduction of two different types of bonds (C=N and C=C bonds) and the suppression of side-reactions, namely aldehyde and/ or olefin reduction, which might take place before the proposed AH–DRA reaction sequence. We envisioned two strategies that may help to tackle those problems. One is the application of a transition metal which could efficiently coordinate and reduce both the imine bond and the olefin bond. The other is the addition of appropriate additives which facilitate the imine formation and the reduction thereafter, and alleviate the inhibition effect from the amine reactant, imine intermediate and the amine product on the catalyst. Fan and co-workers just disclosed an elegant Ir/Ru-catalyzed AH and DRA of quinoline-2-carbaldehydes with anilines[42].

Herein, we describe our efforts toward the integration of AH of olefins and DRA of aldehydes into a one-step cascade reaction. Through the incorporated AH and DRA sequential reactions in one step, the synthetic efficiency of related chiral amino compounds is significantly improved.

## Results

**Establishment of feasibility.** In our previous direct asymmetric reductive amination research, iridium catalyst demonstrated prominent reactivity toward imine reduction[43–45]. So [Ir(cod)Cl]₂ precursor along with (R)-Segphos was initially evaluated in the reaction of α,β-unsaturated aldehyde **1a** and aniline **2a**. Unfortunately only side-product **5** was obtained. Since Brønsted acids have been extensively used in reductive amination to promote the imine formation and imine reduction[46–50], 30 mol% p-toluene-sulfonic acid (TsOH) was added to the reaction. As expected, TsOH boosted the imine formation and imine reduction to afford 21% product **3a** at 64% ee, with 75% intermediate **4** been formed (Table 1, entry 2). When the metal precursor was switched to [Rh

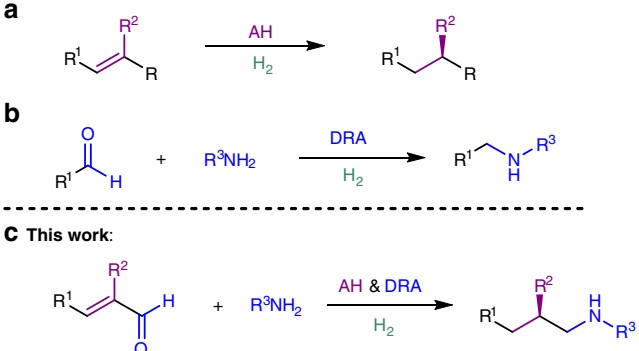

**Fig. 1 The combination of asymmetric hydrogenation (AH) and direct reductive amination (DRA). a** Asymmetric hydrogenation of olefins. **b** Direct reductive amination of aldehydes. **c** This work: the combination of asymmetric hydrogenation and direct reductive amination.

(cod)Cl]₂, TsOH again exercised positive effects on the reaction (entry 4). At the same time, it suppressed the aldehyde direct reduction. From the results, we can see rhodium was more capable of olefin reduction than iridium. In the brief solvents screening (Table 1, entries 4–8), ethyl acetate, which possesses moderate coordinating ability, provided the highest ee. Next, a few Brønsted acids were examined (Table 1, entries 8–11), among which 4-chlorobenzenesulfonic acid facilitated the catalytic system for better reactivity as well as higher stereoselectivity than other acids. Then the influence of the counterion of the cationic Rh complex was studied[51,52]. The results demonstrated that the coordinating anions, such as iodide (Table 1, entry 16) and acetate (Table 1, entry 12), deactivated the reaction, while the noncoordinating anion hexafluoroantimonate improved both the reactivity and the selectivity (Table 1, entry 14). Since the coordinating species had influence on the reaction selectivity, we tried some solvents with coordination ability. It turned out the addition of N,N-dimethyllformamide (DMF) to MeOAc (at the ratio of 1:4) enhanced the reaction ee to 98% (Table 1, entry 17). A control experiment, in which aniline **2a** was not added, afforded **5** in 39% yield with 61% substrate **1a** remaining under the same reaction conditions as in entry 17 (Fig. 2a). The above results demonstrated that the construction of the C–N bond beforehand helped to pave the way for the C=C bond reduction (Fig. 2b).

**Examination of substrate scope.** With the optimal conditions using the rhodium/Segphos catalyst and the additive set in hand, we first explored the scope of the α,β-unsaturated aldehydes in the AH and DRA reactions with aniline **2a** (Table 2). In most cases, the aldehyde substrates could be transformed into the corresponding chiral amino compounds in excellent enantioselectivity and high yields. Regarding the substituted aromatic groups connected to the chiral center (R²) of the products, when the substituents were on para- or meta-positions, the corresponding products were obtained with more than 95% ee and 90% yields, regardless of their electron-donating (**3b–3e**, **3h–3j**) or electron-withdrawing (**3f–3g**, **3k**) properties; but when the substituents were on ortho-positions (**3l–3n**) or as 1-naphthyl group (**3p**), the reaction required higher catalyst loading and/or reaction temperature, probably due to the greater steric hindrance. The additive set and catalytic system also worked well for heteroaromatic substrate **1q** and aliphatic group substituted **1r**. It is worth mentioning that protic groups –OH (**3e**) and –NHAc (**3i**), and reducible group –CN (**3k**) were well-tolerated in the reactions. As for the various –R¹ substrates, the results were similar to that of the –R² substituted ones. As for the limitations

**Table 1 Initial AH & DRA investagation of 2,3-diphenylacrylaldehyde 1a and aniline 2a.[a]**

| Entry | Metal precursor | Solvent | Additive[b] | Yield of 3a (%) | ee (%) |
|---|---|---|---|---|---|
| 1 | [Ir(cod)Cl]$_2$ | THF | – | – | – |
| 2 | [Ir(cod)Cl]$_2$ | THF | TsOH | 21 | 64 |
| 3 | [Rh(cod)Cl]$_2$ | THF | – | – | – |
| 4 | [Rh(cod)Cl]$_2$ | THF | TsOH | 46 | 82 |
| 5 | [Rh(cod)Cl]$_2$ | EtOAc | TsOH | 68 | 75 |
| 6 | [Rh(cod)Cl]$_2$ | CH$_2$Cl$_2$ | TsOH | 91 | 52 |
| 7 | [Rh(cod)Cl]$_2$ | toluene | TsOH | 89 | 63 |
| 8 | [Rh(cod)Cl]$_2$ | MeOH | TsOH | 57 | 18 |
| 9 | [Rh(cod)Cl]$_2$ | EtOAc | PhCO$_2$H | – | – |
| 10 | [Rh(cod)Cl]$_2$ | EtOAc | 4'-ClC$_6$H$_4$SO$_3$H | 85 | 81 |
| 11 | [Rh(cod)Cl]$_2$ | EtOAc | MeSO$_3$H | 52 | 70 |
| 12 | [Rh(cod)Cl]$_2$ | EtOAc | 4'-ClC$_6$H$_4$SO$_3$H NaOAc | 57 | 81 |
| 13 | [Rh(cod)Cl]$_2$ | EtOAc | 4'-ClC$_6$H$_4$SO$_3$H NaBF$_4$ | 72 | 86 |
| 14 | [Rh(cod)Cl]$_2$ | EtOAc | 4'-ClC$_6$H$_4$SO$_3$H NaSbF$_6$ | 90 | 93 |
| 15 | [Rh(cod)Cl]$_2$ | EtOAc | 4'-ClC$_6$H$_4$SO$_3$H NaBArf | 34 | 67 |
| 16 | [Rh(cod)Cl]$_2$ | EtOAc | 4'-ClC$_6$H$_4$SO$_3$H KI | 33 | 60 |
| 17 | [Rh(cod)Cl]$_2$ | MeOAc/DMF 4:1 | 4'-ClC$_6$H$_4$SO$_3$H NaSbF$_6$ | 98 | 98 |

[a]Reaction conditions: [Rh]/Segphos/**1a**/**2a** = 1:1.1:100:100; **1a** 0.1 mmol, solvent 2 mL, 60 °C, 24 h. Segphos = 5,5'-bis(diphenylphosphino)-4,4'-bi-1,3-benzodioxole. MS = molecular sieves, 0.1 g. Yields were isolated yields. Enantiomeric excesses were determined by chiral HPLC.
[b]The amount of added Brønsted acids was 30 mol%. The amount of added counterions was 5 mol%.

**Fig. 2 The order of reduction of C=N and C=C bonds. a** The reduction of **1** without the addition of aniline **2a**. **b** The reduction of **1** with the addition of aniline **2a**.

of the reaction, the catalytic system did not work well when R[1] and R[2] were both alkyl groups (**3zb**), or the α-substituent was switched to β-position (**3zc**). Alkyl amine was also not suitable for this method (**3zd**). The reactions of 2-phenylbut-2-enal (**1ze**) with **2a** did not lead to the desired products. As for 3,4-diphenylbut-3-en-2-one (**1zf**), most of the starting material **1zf** remained untouched.

Then the scope of anilines was explored using the same catalytic system under the optimized conditions. The results are summarized in Table 3. All selected anilines, with substituents at *para-* (**3ab**–**3ad**), *meta-* (**3ae**–**3ag**) and *ortho-* (**3ah**–**3aj**) positions, or having electronic-withdrawing or electronic-donating groups, reductively coupled with the 2,3-diphenylacrylaldehyde substrate **1a** smoothly to afford the desired products in excellent ees and yields. Notably the protic –OH group (**2i**) again tolerated in this transformation. In addition, sterically hindered anilines,

with *ortho*-substituents (**2h**, **2j**) and 1-naphthyl group (**2k**), were all suitable nitrogen sources for the successful conversion of **1a**.

**Practical applications.** To further showcase the utility of this AH and DRA conbination strategy, we next made efforts on the transformations of these N-(2,3-diarylpropyl)aniline products **3**. Tetrahydroquinolines are prevailing natural alkaloids and artificially synthesized molecules, which have found frequent applications in pharmaceutical and agrochemical industry[53–56]. The AH of readily available quinolines is the most convenient and straightforward access to these compounds[6,57]. But the 3-substituted tetrahydroquinolines could not be prepared through this route, since under the typical acidic AH reaction conditions, the 1,4-dihydroquinoline intermediates would undergo isomerization to form 1,2-dihydroquinoline, in which it is hard to control the

## Table 2 Investigation of α,β-unsaturated aldehyde scope.[a]

Reaction scheme: 1 (R¹, CHO, R²) + 2a (PhNH₂) → Rh/(R)-Segphos, MS, 4'-ClC₆H₄SO₃H 30 mol%, NaSbF₆ 5 mol%, MeOAc:DMF, 60 °C, 50 atm H₂, 24 h → 3 (R¹, R², NHPh)

**3a** 98% yield, 98% ee

**3b** 95% yield, 99% ee

**3c** 97% yield, 97% ee

**3d** 94% yield, 97% ee

**3e** 92% yield, 97% ee

**3f** 98% yield, 97% ee

**3g** 98% yield, 97% ee

**3h** 98% yield, 98% ee

**3i** 91% yield, 95% ee

**3j** 97% yield, 98% ee

**3k** 94% yield, 95% ee

**3l**[b] 94% yield, 96% ee

**3m**[c] 88% yield, 97% ee

**3n**[c,d] 92% yield, 96% ee

**3o** 95% yield, 98% ee

**3p**[b] 95% yield, 96% ee

**3q** 95% yield, 95% ee

**3r** 80% yield, 94% ee

**3s** 92% yield, 99% ee

**3t** 97% yield, 98% ee

**3u** 97% yield, 96% ee

**3v** 98% yield, 97% ee

**3w** 98% yield, 95% ee

**3x** 93% yield, 91% ee

**3y**[c,d] 63% yield, 94% ee

**3z**[c,e] 91% yield, 95% ee

**3za** 93% yield, 95% ee

Limitations:

**3zb** 56% yield, 30% ee (44% yield)

**3zc** 34% yield, <5% ee

**3zd** 70% yield, 26% ee

**3ze** <10% yield

**3zf** <5% yield

[a]Reaction conditions: [Rh]/Segphos/**1**/**2a** = 1:1.1:100:100; **1** 0.3 mmol, solvent 4 mL with the MeOAc/DMF ratio at 4:1, 60 °C, 24 h. MS = molecular sieves, 0.3 g. Yields were isolated yields. Enantiomeric excesses were determined by chiral HPLC. PMP = 4-methoxyphenyl.
[b]2 mol% Rh–(R)-DM-Segphos was used. No NaSbF₆ was added. DM-Segphos = 5,5'-Bis[di(3,5-xylyl)phosphino]-4,4'-bi-1,3-benzodioxole.
[c]2 mol% catalyst was used.
[d]The amine source was p-anisidine. The reaction temperature was 70 °C.
[e]The amine source was p-anisidine.

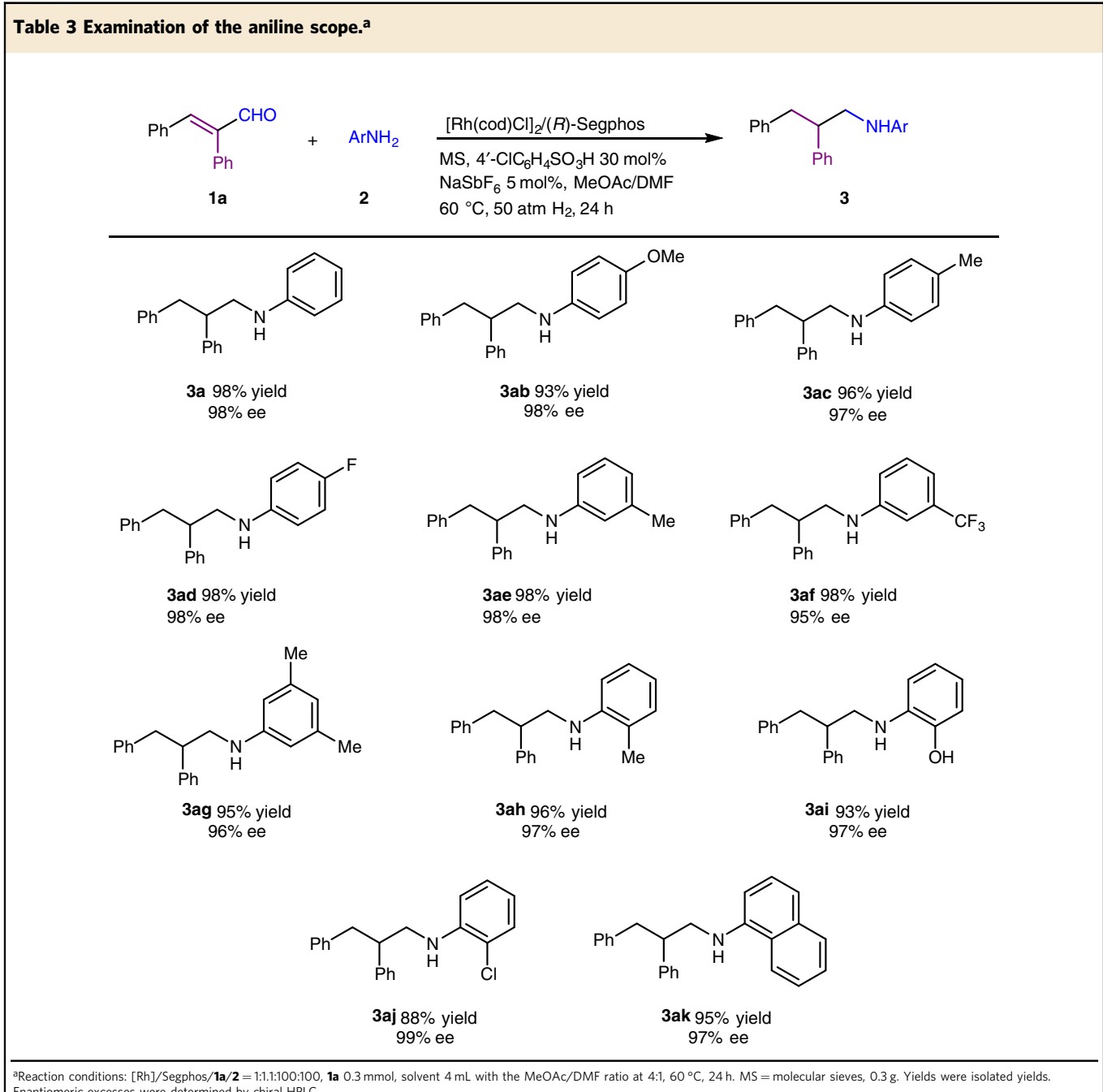

**Table 3 Examination of the aniline scope.[a]**

**3a** 98% yield
98% ee

**3ab** 93% yield
98% ee

**3ac** 96% yield
97% ee

**3ad** 98% yield
98% ee

**3ae** 98% yield
98% ee

**3af** 98% yield
95% ee

**3ag** 95% yield
96% ee

**3ah** 96% yield
97% ee

**3ai** 93% yield
97% ee

**3aj** 88% yield
99% ee

**3ak** 95% yield
97% ee

[a]Reaction conditions: [Rh]/Segphos/**1a**/**2** = 1:1.1:100:100, **1a** 0.3 mmol, solvent 4 mL with the MeOAc/DMF ratio at 4:1, 60 °C, 24 h. MS = molecular sieves, 0.3 g. Yields were isolated yields. Enantiomeric excesses were determined by chiral HPLC.

stereoselectivity[6,58]. Using our methodology, product **3y** from the 3-(2-chlorophenyl)-2-phenylacrylaldehyde **1y** substrate could easily been converted into the chiral 3-phenyl-tetrahydroquinoline product **6** via the Buchwald–Hartwig cross-coupling reaction (Fig. 3a), in which the ee value was not affected[59–61]. Similarly, product **3n** from the 2-(2-chlorophenyl)-3-phenylacrylaldehyde **1n** substrate could be cyclized to form the 3-benzylindoline alkaloid **7** (Fig. 3b).

## Discussion

In summary, we have successfully integrated in one-pot two efficient reactions, AH and DRA, which share the common reductant, namely hydrogen gas. Catalyzed by the rhodium-Segphos complex, the DRA of aldehydes and the AH of prochiral olefins took place sequentially to afford the chiral amino compounds. The rhodium catalyst precursor was capable of

performing the reduction of two different types of bonds. The addition of 30 mol% of 4-chlorobenzenesulfonic acid facilitated the imine formation, imine reduction and the C=C bond reduction. Noncoordinated counterion of the cationic rhodium complex hexafluoroantimonate improved both the reaction reactivity and the stereoselectivity. With our protocol, useful chiral amino compounds can be synthesized in a more convenient and effective manner.

## Methods

**General procedure for asymmetric reductive amination**. In a nitrogen-filled glovebox, [Rh(cod)Cl]$_2$ (5 μmol) and (R)-Segphos (11 μmol) was dissolved in anhydrous CH$_3$COOCH$_3$ (1.0 mL), stirred for 20 min, and equally divided into 10 vials charged with aldehyde (0.1 mmol) and aniline (0.1 mmol) in anhydrous CH$_3$COOCH$_3$ solution (0.5 mL). Then 4-Cl–PhSO$_3$H (0.3 equiv.) and NaSbF$_6$ (0.05 equiv) were added and the total solution was made to 2.0 mL (MeOAc:DMF = 4:1)

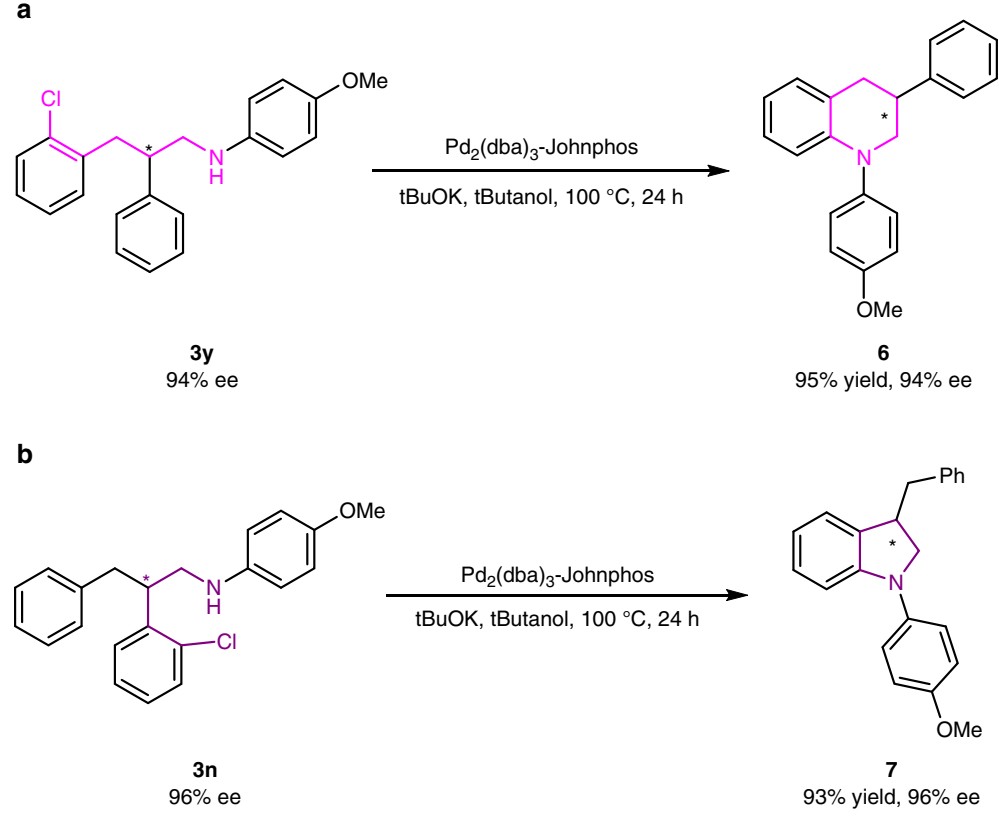

**Fig. 3 Synthesis of a 3-tetrahydroquinoline and a 3-indoline. a** The transformation of **3y** for the synthesis of 3-phenyl-tetrahydroquinoline **6**. **b** The transformation of **3n** for the synthesis of 2-benzyl-indoline **7**.

for each vial. The resulting vials were transferred to an autoclave, which was charged with 50 atm of $H_2$, and stirred at 60 °C for 24 h. The hydrogen gas was released slowly and the solution was quenched with aqueous sodium bicarbonate solution. The organic phase was concentrated and passed through a short column of silica gel to remove the metal complex to give the crude products, which were purified by column chromatography and then analyzed by chiral HPLC to determine the enantiomeric excesses.

## Data availability

The experimental procedure and characterization data of new compounds are available within the Supplementary Information. Any further relevant data are available from the authors upon reasonable request.

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

## Acknowledgements

Financial support from the National Natural Science Foundation of China (21772155) is gratefully acknowledged.

## Author contributions

S.Y. established the reaction conditions. G.G. and C.L. prepared the substrates. S.Y., L.Wang., L.Wan., H.H. and H.G. expanded the substrate scope. M.C. conceived and supervised the project and wrote the manuscript. All the authors discussed the results and commented on the manuscript.

## Competing interests
The authors declare no competing interests.
