## [Peer Review File · Nature Communications]

Reviewers' comments:

Reviewer #1 (Remarks to the Author):

Disclosure: By chance I reviewed a previous version of this manuscript for another journal, where I recommend acceptance after revisions. In this revised manuscript submitted to Nature Communication, the authors addressed nearly all my comments (except one, see below) which from my point of view really improved the manuscript and therefore my comments here are quite positive.

A system for the combination of an asymmetric olefin hydrogenation with a reductive amination of an aldehyde on α,β -unsaturated aldehydes using H₂ and a single Rh-Segphos catalyst for both hydrogenations is presented in this manuscript.

Despite the asymmetric hydrogenation of the C=C-bond of prochiral α,β -unsaturated aldehydes to the corresponding chiral aldehyde is well described and carried out on a significant industrial scale (e.g. in the BASF-process to produce menthol based on citral) and also the reductive amination of aldehydes using H₂ and metal catalysts (see refs 7 in the manuscript), the combination of the two steps using a single metal-catalyst is quite elegant and the yields as well as the ee's achieved with the investigated substrates is very good. It seems that adding enough acid as well as molecular sieve to achieve a fast and complete imine formation to suppress hydrogenation of the carbonyl-group to the alcohol is the trick. The authors could also show, that it seems to be crucial, that the combination of both steps is essential in this system for good ee's (see Eq. 1).

The substrate-scope is from my point of view good, but the authors could also show some limitations of this system, which helps the reader (entries 3zc, 3zb, 3zd in table 2), if one wants to adapt this system in an own synthesis.

It's a quite interesting system, which allows an elegant access to chiral amino compounds and is worth to be published in Nature Communications.

I just have one comment, to be addressed: What happens, when a prochiral α,β -unsaturated ketone is used instead of an aldehyde? Will then also the imine-reduction be enantioselective with this catalyst? This would be very interesting, if one can introduce two stereocenters selectively with the same catalyst when performing the reduction/amination of an α,β -unsaturated ketone. If not, it should at least be mentioned, that this is not working. For me it seems obvious also to try this.

Reviewer #2 (Remarks to the Author):

Chang, Geng and co-workers have developed an interesting relay reduction chemistry involving sequential asymmetric hydrogenation of an alkene and reductive amination of an aldehyde group. Although either single step is well studied, the combination of two reductive processes with a singular metal catalyst is challenging and thus highly attracting. Usually, such combination outputs complex mixtures. Considering the broad scope, high yields, excellent enantiocontrol as well as easy access of substrates, this method represents a useful and practical method to achieve β -chiral amines which are important chiral units in numerous natural and biological compounds. Overall, this work is worthy to be published in this journal after minor revision.

There are some suggestions to improve the quality of this work.

1. the writing should be polished as the draft is not fluent enough.
2. the SI is overall well prepared and the reaction procedures are easy to follow. There is a mistake that all indicated retention times of each enantioenriched compound are adapted from the spectra of the corresponding racemic compound, which should be corrected.

Reviewer #3 (Remarks to the Author):

This paper describes the novel combined action of catalytic asymmetric hydrogenation and reductive amination in one step. Efficient conditions have been found to give high enantioselectivity in a number of cases, and the experimental work has been conducted to an acceptably high standard. There are flaws both in the construction of the MS and the choice of experiments that militate against acceptance in its present form and without further experimental input. A properly modified revised version could well be acceptable.

The first problem lies in the Introduction. The background to asymmetric hydrogenation and reductive amination are well known to the catalysis community, and the first two pages could be replaced by a short discussion with appropriate background and a Figure that makes the in situ production of the imine prior to hydrogenation clearer (see the experiment in unlabelled Figure page 4).

A more serious problem is that the scope and limitations of their reduction are not well defined; almost all the profuse examples have aromatic substitution at all three sites of variation in the imine substrate. With both substituents on the C=C double bond as alkyl the reaction is inefficient; what about one alkyl on the terminal carbon, and other structural variations that inform the reader of how useful this procedure might be to the community.

In the SI, be sure to include references where the substrate and/or the product is known, unless the source is commercial. An important control experiment is to make sure that the ee is not influenced by chromatographic purification (cf Kagan's work in JOC) by reporting data for the crude product in several cases.

For reviewer 1:

- 1) I just have one comment, to be addressed: What happens, when a prochiral α,β -unsaturated ketone is used instead of a aldehyde? Will then also be the imine-reduction be enantioselective with this catalyst? This would be very interesting, if one can introduce two stereocenters selectively with the same catalyst when performing the reduction/amination of a α,β -unsaturated ketone. If not, it should at least be mentioned, that this is not working. For me it seems obvious also to try this.

Thanks for the suggestion. We have carried out the reaction of α,β -unsaturated ketone 3,4-diphenylbut-3-en-2-one (**1zf**) with aniline **2a**. Unfortunately most of the starting material **1zf** remained untouched even after the reaction temperature was elevated to 80 °C. So it requires other reaction conditions for the α,β -unsaturated ketone substrates to react with **2a**.

For reviewer 2:

- 1) The writing should be polished as the draft is not fluent enough;

Thanks. We have rewritten some sentences.

- 2) the SI is overall well prepared and the reaction procedures are easy to follow. There is mistake that all indicated retention time of each enantioenriched compound is adapted from the spectra of the corresponding racemic compound, which should be corrected.

Thanks. The retention time of the enantiomers has been changed to the corresponding time of the enantioenriched compounds.

For reviewer 3:

- 1) The first problem lies in the Introduction. The background to asymmetric hydrogenation and reductive amination are well known to the catalysis community, and the first two pages could be replaced by a short discussion with appropriate background and a Figure that makes the in situ production of the imine prior to hydrogenation clearer (see the experiment in unlabelled Figure page 4);

Thanks. We have rewritten the introduction and added Figure 2 to elucidate the order of the reductive amination and the olefin reduction. Considering *Nature Communications* has a broad readership, we didn't do major change to the

asymmetric hydrogenation and reductive amination introduction, but only made minor modifications.

- 2) A more serious problem is that the scope and limitations of their reduction are not well defined; almost all the profuse examples have aromatic substitution at all three sites of variation in the imine substrate. With both substituents on the C=C double bond as alkyl the reaction is inefficient; what about one alkyl on the terminal carbon, and other structural variations that inform the reader of how useful this procedure might be to the community.

Thanks for the suggestion. We have included 4 substrates that have alkyl substituents in Table 2, **1r**, **1zb**, **1zc** and **1ze**. For α -Me substrate **1r**, the reaction went smoothly to afford product **3r** in high ee. For α -Me- β -Et substrate **1zb** and β -Me- β -Ph substrate **1zc**, the reaction provided **3zb** and **3zc** as the major products, but with low ee. For α -Ph- β -Me substrate **1ze**, the reaction provided no major product.

- 3) In the SI, be sure to include references where the substrate and/or the product is known, unless the source is commercial. An important control experiment is to make sure that the ee is not influenced by chromatographic purification (cf Kagan's work in JOC) by reporting data for the crude product in several cases.

Thanks for the suggestion. We have added references for known substrates and products, and added the characterization spectra (^1H NMR, ^{13}C NMR and HRMS) for unknown substrates and products. We further purified products **3d**, **3k**, **3n**, **3s**, **3y**, **3z** and **3ai**, and found the ee values were the same with the values before their purification.

REVIEWERS' COMMENTS:

Reviewer #3 (Remarks to the Author):

The authors have successfully incorporated the major comments of referees, and subject to editorial scrutiny the MS is suitable for publication now.